# Does CSR Information Disclosure Improve Investment Efficiency? The Moderating Role of Analyst Attention

Zhen Li [1], Shenglan Li [2], Zhuoyu Huo [2], Yuxia Liu [2] and Hua Zhang [3,*]

1   School of Accounting, Hunan University of Finance and Economics, Changsha 410205, China; lizhen@hufe.edu.cn
2   School of Business, Central South University, Changsha 410017, China; 211612238@csu.edu.cn (S.L.); 221612240@csu.edu.cn (Z.H.); lyxlxh08020714@163.com (Y.L.)
3   Institute of Innovation and Entrepreneurship, Loughborough University London, London E20 3BS, UK
*   Correspondence: h.d.zhang@lboro.ac.uk

**Abstract:** Since 2009, the China Securities Regulatory Commission has begun to require listed firms on the specified boards to disclose their corporate social responsibility and encouraged others to report corporate social responsibility voluntarily. Based on the data of domestic A-share listed companies from 2013 to 2019, this paper studies the relationship between corporate social responsibility information disclosure and corporate investment efficiency and the role of analysts in moderating the relationship. The empirical results show that the social responsibility information disclosed under China's mandatory guidance has a positive effect on alleviating information asymmetry and improving investment efficiency, and this role becomes even more crucial when the external information environment fails to meet market demands. Overall, our findings suggest the important role of corporate social responsibility information disclosure in guiding investment behavior and improving investment efficiency, especially for those companies with low analyst attention. This article expands the research perspective on social responsibility information disclosure and investment efficiency. Furthermore, our research contributes to promoting corporate social responsibility and facilitating sustainable development.

**Keywords:** corporate social responsibility; information disclosure; analyst attention; investment efficiency

## 1. Introduction

With the concept of coordination and green civilization becoming the socialist ideology with Chinese characteristics in the new era, the degree of concern for corporate social responsibility (CSR) has continued to increase as a hot spot of public concern. Enterprises have gradually recognized the importance of implementing their social responsibilities such as biodiversity protection and climate governance. As important non-financial information, social responsibility information is regarded as a signal of management ethics and integrity [1], which can enrich the communication channels between companies and investors and, simultaneously, improve the quality of information obtained by investors [2], leading to the fact that CSR disclosure as a core social responsibility activity has been taken seriously. Since December 2008, the Shenzhen and Shanghai stock exchanges have issued public announcements, the Chinese capital market has officially entered the "CSR forced disclosure era", and the scale and quality of CSR information disclosure have reached a new level, with a dramatic increase in the number of CSR reports released during 2009–2017.

However, there is still much controversy in the research on the advantages and disadvantages of the CSR mandatory disclosure policy [3]. While CSR disclosure improves the level of non-financial information disclosure and weakens information asymmetry, it generates direct and indirect costs that need to be borne by equity stakeholders. On the one hand, the improvement of CSR information disclosure level helps enterprises to send

positive signals to the capital market, establish a highly responsible corporate image [4], reduce financing costs [5], improve analysts' forecasting accuracy and tracking quantity [6], reduce litigation risks [7], and enhance corporate reputation [8]. On the other hand, CSR information disclosure may occupy enterprise resources, increase disclosure costs, and damage enterprise performance. In short, if CSR is regarded as a value-added behavior and investors respond strongly, CSR information disclosure can bring positive benefits to enterprises [9]. When social responsibility activities to meet the needs of stakeholders are considered to be a waste of resources, corporate value is reduced.

The existing literature pays more attention to the economic consequences of CSR information disclosure [10], and the mechanism between information disclosure and firm value has been extensively studied [11,12]. Being heavily influenced by the mandatory disclosure policy, investment efficiency focuses on corporate strategy and sustainable development and has long been the focus of corporate various stakeholders. However, as an important factor affecting firm value, there is little literature on the impact of CSR information disclosure on the investment efficiency of enterprises in the context of mandatory disclosure policy. Meanwhile, very few studies have elaborated on the relationship between CSR disclosure and corporate investment efficiency, especially under the mandatory disclosure policy of CSR information. According to Ref. [13], which viewed CSR as an external behavior, whether CSR information disclosure can have a positive impact on enterprises depends on investors' perception of CSR activities. Such a situation has suggested the essence of studying the relationship between social responsibility information disclosure and investment efficiency under the background of the mandatory disclosure policy.

As an important role in the operation of the capital market, financial analysts have the functions of information intermediary and external supervision, which plays an indispensable role in promoting listed companies to improve investment efficiency [2]. By excavating enterprise management information and financial information, analysts publish enterprise research reports to investors, improve the transparency of corporate information, reduce agency costs caused by the separation of powers, and alleviate information asymmetry, which is an important reason for inefficient investment [14], whereas the normative development time of analysts in the capital market is still short, and its impact on investment efficiency is worthy of further investigation. At present, academics lack research on the relationship between analysts and investment efficiency. CSR information as an important source of information for analysts enables analysts to play an important role in the external supervision of the information disclosed by listed companies [15], which suggests the necessity of integrating analysts into the framework of studying the impact of CSR information disclosure on investment efficiency so as to fully understand the influential factors of investment efficiency. This paper will focus on whether CSR information can provide effective marginal information increments to influence investment efficiency and, simultaneously, can make a useful supplement to the existing research. It can also provide guidance for the disclosure of CSR information for enterprises with different analysts' attention and also provide a reference for the government departments to improve the CSR information disclosure system. Our research also provides a reference for the research of the relationship between analysts' attention and CSR information disclosure.

This paper adopts stakeholder theory [16], information asymmetry theory [17], and organizational legitimacy theory [18] and refers to Ref. [19]'s method to distinguish investment scenarios. It aims to analyze the influence of CSR information disclosure on investment efficiency in both the overall sample and subsamples that distinguish investment scenarios. Additionally, it examines the impact of analysts' attention on the relationship between CSR information disclosure and investment efficiency and explores whether this impact differs significantly. The findings emphasize the crucial role of CSR information disclosure in guiding investment behavior and improving investment efficiency, especially for companies with low analyst attention.

Whilst previous research mostly explores the influencing factors of investment efficiency from the internal factors of the enterprise [3], very few studies pay attention to the

impact of capital market analysts on the relationship between social responsibility information disclosure and investment efficiency, and this paper provides new evidence for the research of analysts' attention on the behavior and value of micro-enterprises, which may therefore contribute to enriching the existing literature on CSR disclosure and investment efficiency. This paper also extends the contemporary literature regarding financial analysts and CSR disclosure [20] by addressing the importance of analysts' attention in shaping corporate investment efficiency, which may contribute to generating research agendas in CSR and financial analysts in emerging economies.

This paper is structured as follows: Section 2 presents the contemporary literature on CSR disclosure and investment efficiency so as to lead to hypothesis development; Section 3 is the research design, with the empirical analysis in the next Section 4, followed by the robustness text in Section 5. The conclusion and the prospect are presented in the last Section 6, with insights on potential contributions and implications.

## 2. Literature Review and Hypothesis Development

### 2.1. CSR Information Disclosure and Investment Efficiency

Previous studies propose that CSR information disclosure affects financial performance and corporate value [11,12], and the increase in corporate value has been regarded [21] to depend fundamentally on the improvement of investment efficiency. Investment activity is the core of corporate financial activities and is one of the most dynamic and long-term influential resource allocation behaviors of enterprises [3]. The contemporary literature has widely discussed the relationship between CSR information disclosure and investment efficiency from the perspectives of stakeholder theory, information asymmetry theory, and signal transmission theory [22]. Stakeholder theorists [16] argue that CSR disclosure can effectively improve the efficiency of contract formation among stakeholders and reduce contract costs. Corporate decision making, especially long-term strategic decision making, tends to consider the interests of non-equity stakeholders to obtain long-term stable and sustainable resource input and support [23]. CSR activities are essentially the management of all stakeholders, and CSR information disclosure by enterprises should be seen as an important part of the strategic plan for stakeholder relations [24]. The diversified common governance structure composed of stakeholders can encourage each contract entity to effectively supervise the operation of enterprises, thereby improving corporate governance performance and improving investment efficiency [2]. Therefore, the satisfaction of stakeholder requirements is an important motivation for companies to disclose CSR information [25,26].

In addition, information asymmetry theory [5] holds that there are differences in the mastery of transaction information by all kinds of market participants in economic activities, and more delicate participants are often in a more favorable position. Previous studies have found that the most fundamental reason for the disturbance of corporate investment efficiency is information asymmetry. The disclosure of non-financial information is helpful for reducing the degree of information asymmetry between enterprises and investors [27], as well as between stakeholders and operators, and effectively transmitting valuable incremental information to the capital market [28,29]. The former is conducive to reducing corporate finance costs, enhancing corporate reputation in the capital market, which in turn helps attract investors, enhances investor confidence, and alleviates the underinvestment of enterprises [30], whereas the latter can promote the effective supervision of the operators by stakeholders and avoid the operators from making inefficient investment decisions that are not conducive to the long-term development of the enterprise in order to enhance their reputation and salary, such as blindly expanding the investment, thus inhibiting excessive investment. Therefore, CSR information disclosed by the company is an important way to reduce the degree of information asymmetry and improve investment efficiency [31].

Organizational legitimacy [18] refers to how an entity's actions are required and appropriate in the system of norms, values, beliefs, and interpretations constructed by society. In other words, organizational legitimacy is the recognition of the organization by

stakeholders or bystanders. Legitimacy theory argues that the company's disclosure of CSR information is a response to environmental factors such as public pressure, and its purpose is to prove that the values and goals pursued by the company are consistent with the social group through legality management [32,33]. Neo-institutional theory [34] holds that behavioral imitation between enterprises is an important way to obtain "rationality" and reduce uncertainty [35]. Large-scale or high-performing companies will send reasonable signals to other companies to imitate as a way to enhance legitimacy [19,31], and the investment efficiency of enterprises is thus improved. As an important activity to enhance corporate image, CSR information disclosure is an important way to gain legitimacy and enhance the confidence of stakeholders and society in enterprises to improve investment efficiency. Therefore, this paper proposes the following assumption:

**Hypothesis 1 (H1).** *Companies that conduct CSR information disclosure have higher investment efficiency.*

### 2.2. Financial Analyst Attention

The existing literature shows that enterprise information disclosure affects the decision-making process of analysts. As the information bridge between enterprises and stakeholders, enterprise analysts are very concerned about the information disclosure of enterprises, and according to the trend direction of information, this attention is asymmetrical. Especially when the trend of financial information is negative, the non-financial information is more concerned, and analysts use non-financial performance information to conduct corporate valuation, which will have a certain guiding effect on investors' subsequent investment decisions [36]. The contemporary literature on the relationship between analyst attention and investment efficiency based on information asymmetry theory and stakeholder theory has identified that securities analysts can play a role in reducing information asymmetry [37] and supervising company management [38].

Information asymmetry perspectives [17] believe that companies with higher analysts' attention have a lower degree of information asymmetry with external investors and more efficient communications with external markets [39,40]. Investors can investigate more clearly the operating performance of managers and play a supervisory role, which is conducive to promoting managers to improve their decision-making levels and thus improve their investment efficiency [41]. In addition, the analysts' attention and interpretation of corporate information helps investors to identify and interpret investment information, evaluate the quality and value of corporate CSR disclosure, and avoid being manipulated and confused by the packaging of enterprises. Therefore, the higher the analyst's attention, the better the external information environment of the enterprise, and the more favorable the information mining and transmission [15]. Meanwhile, corporate governance theorists believe that external financial analysts are one of the most important external oversight bodies of the company's management [42]. Analysts' attention has a supervisory effect on companies, and their continuous attention will reduce the agency problem [43] in enterprise investment decision making, which may cause managers to avoid expanding investment for personal gain and improve investment efficiency.

Due to the existence of information asymmetry, external investors cannot realistically estimate the value of investment projects, resulting in underinvestment, or management selectively discloses information that is beneficial to them and increases market investors' expectations for investment projects, leading to the problem of excessive investment. When CSR is disclosed by enterprises with analysts' concerns, CSR information disclosure provides additional information for analysts' forecasts. The improvement of CSR disclosure level improves the accuracy of analysts' forecasts, and analysts can effectively transmit real investment information to market investors, provide useful information for investors, reduce the impact of information asymmetry on external investors, and thus improve investment efficiency [20]. Therefore, we assume that analysts' attention plays a regulatory role in the relationship between CSR information disclosure and investment efficiency.

Moreover, analysts' attention partly overlaps with the market effect and supervision role of CSR information disclosure [44]. The impact of CSR information disclosure on investment efficiency may be affected by differences in analysts' attention. It is generally believed that the higher the analysts' attention, the faster the information is excavated and disseminated [8], and the more transparent the information environment of the company [45], the greater the supervisory role of the company. Accordingly, the extent of their demand and dependence on corporate CSR disclosure is also weakened, and the supervision effect is also weakened. Conversely, if the analysts' attention is lower, and the information environment is less transparent, the information that the company actively discloses will become the main source of information needed by investors [46]. In that case, the disclosure of CSR information or the improvement of information quality will increase the information content of the capital market and attract more attention from investors, and the more obvious the signal transmission effect will be. This is more convenient for enterprise impression management [7], which makes it easier for enterprises to disclose CSR to establish a good public image, win the favor of investors, and improve the investment efficiency of enterprises. Based on the above discussion, we argue that CSR information disclosure will be more obvious to the company's investment efficiency when the analysts' attention is low. The following hypothesis is therefore proposed, which supplements H1 to build up the theoretical framework of this paper (see Figure 1):

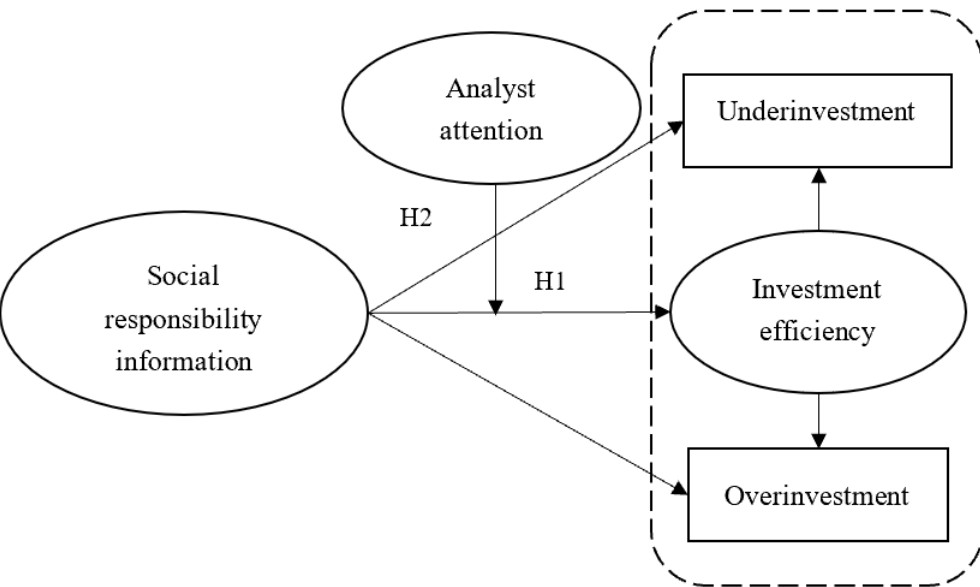

**Figure 1.** Research framework.

**Hypothesis 2 (H2).** *Among companies with low analysts' attention, CSR disclosure has a stronger effect on improving the investment efficiency of enterprises.*

### 3. Materials and Methods

*3.1. Model Construction*

We construct test models (1) and (2), equation (1) is to test hypothesis 1 and (2) is to test hypothesis 2. The test model in this paper is set to the following form:

$$\text{Invest\_effect}_{i,t+1} = \beta_0 + \beta_1 CSR_{i,t} + \beta_2 ANONO_{i,t} + \gamma controls_{i,t} + \sum INDU + \sum YEAR + \varepsilon_{i,t+1} \tag{1}$$

$$\text{Invest\_effect}_{i,t+1} = \beta_0 + \beta_1 CSR_{i,t} + \beta_2 ANONO_{i,t} + \beta_3 CSR_{i,t} * ANONO_{i,t} + \gamma controls_{i,t} + \sum INDU + \sum YEAR + \varepsilon_{i,t+1} \tag{2}$$

### 3.2. Variable Design

#### 3.2.1. Social Responsibility Information Disclosure

Social responsibility information disclosure is measured by the dummy variable CSR. If the listed company discloses social responsibility information, the value is 1; otherwise, it is 0. The data is derived from the Runling Global Ratings Agency (RKS) database.

#### 3.2.2. Analyst Attention

Existing research usually uses the number of analysts who make profit forecasts and investment rating opinions for target listed companies during the year as a proxy variable to measure analysts' attention. This paper also uses this index to measure the degree of analyst attention (ANONO). Drawing on the practices of Lang and Lundholm (1996) [37], this paper calculates the level of analyst attention based on the number of analysts who publish investment rating opinions and earnings forecasts for each listed company every year. Using the median level of analysts' attention as the dividing standard, the sample above the median level is divided into high analyst attention groups; otherwise, it is classified into the low analyst attention group.

#### 3.2.3. Investment Efficiency

This paper adopts the residual measure method of the Richardson model [19], which divides the investment of the enterprise into two parts; one part is the maintenance investment required for the normal operation of the enterprise, and the other part is the enterprise to increase the new project. The residual of the investment model is used to quantify the investment efficiency of the enterprise. The calculation process of investment efficiency is as follows: first, Equation (3) is used to perform regression industry by industry year by year, and then the absolute value of residual is multiplied by 100 as the proxy variable for the non-efficiency investment amount. Among them, the sample with the positive original regression residual is divided into the overinvestment situation, and the negative residual is the underinvestment situation.

The regression model of investment efficiency residuals is:

$$Invest_{i,t} = \beta_0 + \beta_1 Growth_{i,t-1} + \beta_2 LEV_{i,t-1} + \beta_3 Cash_{i,t-1} + \beta_4 AR_{i,t-1} + \beta_5 SIZE_{i,t-1} + \beta_6 AGE_{i,t-1}$$
$$+ \beta_7 Invest_{i,t-1} + \sum Year + \sum Ind \qquad (3)$$

Invest is the company's current total investment, which is the "cash paid for the purchase and construction of fixed assets, intangible assets and other long-term assets" in the t year cash flow statement minus the "net cash received from disposal of fixed assets, intangible assets and other assets" then divided by the total initial assets; Growth is the growth opportunity, using the lag of the first-stage operating income growth rate to represent it; LEV is the asset–liability ratio, which is the total liabilities divided by the total assets at the end of the period; Cash is free cash flow, which is equal to the net cash flow generated from current business activities divided by total assets at the end of the period; AR is the individual stock return rate, which is measured by adjusting the basic earnings per share last year; Size is the size of the enterprise, measured by the natural logarithm of the total assets; and the year and industry are control variables.

#### 3.2.4. Other Control Variables

Referring to the results of Zhong and Xu (2017) [3] and Chen et al. (2011) [22], the control variables selected in this paper are shown in Table 1. In addition, Table 1 summarizes the definition and measurement of all variables.

### 3.3. Data Collection

This paper selects the A-share listed companies traded in Shenzhen and Shanghai stock markets in 2013–2019 as the initial research sample and performs the following processing on the original data: (1) excluding the sample observations of listed companies

in the financial industry, compared with other industries, as its accounting statements are special; (2) excluding the sample observations of listed companies with incomplete financial data and corporate governance structure data; and (3) excluding ST and PT listed company sample observations, as these companies have abnormal financial data or have been losing money for more than two years. Incorporating them into the research sample will affect the research conclusions. Finally, as shown in Table 2, the total number of sample observations was 12,219, of which 3661 were samples disclosing CSR information, of which the mandatory disclosure ratio was approximately 62% (2278). In order to avoid the influence of extreme values, this paper performs a 5% level tailing treatment on all continuous variables. The measurement of CSR information disclosure comes from the third-party rating agency Runling Global (RKS) database, analyst data, stock price data, and company financial data mainly from the CSMAR database.

**Table 1.** Variable Definition Table.

| | Variable Symbol | Variable Name | Calculation Method |
|---|---|---|---|
| Main variable | Invest_effect | Investment efficiency | Multiply the absolute value of the residual of Equation (1) by 100 |
| | Under_Invest | Underinvestment | Equation (1). The absolute value of the residual greater than 0 multiplied by 100 |
| | Over_Invest | Overinvestment | Equation (1). The absolute value of the residual less than 0 is multiplied by 100 |
| | CSR | Social responsibility information disclosure | Whether to disclose social responsibility information. If yes, take 1; otherwise, take 0 |
| | ANANO | Analyst attention | Tracking the number of securities analysts in the company during the year |
| Control variable | Age | Company age | The natural logarithm of the difference between the company's listing year and the current year |
| | LEV | Assets and liabilities | Ratio of total liabilities to total assets |
| | ROA | Financial Performance | Operating profit at the beginning of the total assets |
| | FRQ | Financial information quality | The absolute value of the residuals of the KASZ model (Kasznik, 1999) [32] is used to measure the quality of the company's financial information: $\frac{TAC_j}{A_j} = \alpha_{i,t} \frac{1}{A_j} + \beta_{1i,t} \frac{\Delta ADJREV_j}{A_j} + \beta_{2i,t} \frac{PPE_j}{A_j} + \beta_{3i,t} \frac{\Delta CFO_j}{A_j} + \varepsilon_j$ TAC is the change in non-current assets minus the change in current liabilities plus the change in short-term borrowings, A is the total assets at the beginning of the period, $\Delta ADJREV$ is the current sales income change, PPE is the current fixed assets, and $\Delta CFO$ is the current operating cash flow fluctuations. |
| | Slack | Resource slack | The ratio of the company's cash and cash equivalents to fixed assets |
| | INS | Institutional investor holdings | Institutional investors holding shares/total number of shares |
| | TOP | The shareholding ratio of the largest shareholder | Number of shares held by the largest shareholder/total number of shares |
| | INDRATIO | Ratio of independent directors | Number of independent directors/board size |
| | So_priv | Whether state-owned holding | State-owned holdings take 1; otherwise, take 0 |
| | MAOW | Executive shareholding ratio | Executive shareholding/total number of shares |

**Table 2.** Sample Screening.

| Screening Process | 2013 | 2014 | 2015 | 2016 | 2017 | 2018 | 2019 | Total |
|---|---|---|---|---|---|---|---|---|
| Sample of A-share listed companies that disclosed investment amount in 2013–2019 | 2341 | 2470 | 2515 | 2632 | 2823 | 3118 | 3488 | 19,387 |
| Less: Sample of listed companies with missing investment efficiency model data | 256 | 385 | 191 | 174 | 319 | 499 | 679 | 2503 |
| Sample of listed companies in the financial and insurance industry | 135 | 135 | 143 | 144 | 144 | 146 | 148 | 995 |
| ST, PT listed company sample | 10 | 17 | 14 | 21 | 13 | 11 | 5 | 91 |
| Sample of listed companies without analyst forecasts | 339 | 324 | 480 | 619 | 583 | 512 | 455 | 3312 |
| Sample of listed companies with missing financial data | 112 | 38 | 27 | 64 | 20 | 4 | 2 | 267 |
| Final sample | 1489 | 1571 | 1660 | 1610 | 1744 | 1946 | 2199 | 12,219 |

## 4. Empirical Analysis

### 4.1. Descriptive Statistics

Descriptive statistics were performed on variables using stata 12.0. The results are shown in Table 3.

**Table 3.** Descriptive statistics for variables.

| Variable Name | Number of Samples | Mean | Standard Deviation | Minimum Value | Maximum | 25% Quantile | Median | 75% Quantile |
|---|---|---|---|---|---|---|---|---|
| Invest_Effect | 12,219 | 4.053 | 7.758 | 0.0003 | 404.7 | 1.240 | 2.670 | 4.960 |
| Over_Invest | 5,048 | 4.900 | 11.34 | 0.0002 | 404.7 | 1.041 | 2.565 | 5.575 |
| Under_invest | 7,171 | 3.456 | 3.353 | 0.001 | 81.93 | 1.373 | 2.733 | 4.712 |
| CSR | 12,219 | 0.300 | 0.458 | 0 | 1 | 0 | 0 | 1 |
| ANONO | 12,219 | 12.28 | 10.770 | 1 | 73 | 4 | 9 | 18 |
| Age | 12,219 | 15.56 | 5.853 | 7 | 24 | 12 | 15 | 19 |
| LEV | 12,219 | 0.425 | 0.232 | 0.008 | 8.256 | 0.250 | 0.417 | 0.591 |
| ROA | 12,219 | 0.078 | 0.367 | −0.604 | 30.94 | 0.022 | 0.055 | 0.099 |
| FRQ | 12,219 | 0.279 | 0.930 | 0 | 54.25 | 0.084 | 0.181 | 0.329 |
| Slack | 12,219 | 5.069 | 34.97 | −0.435 | 1671 | 0.325 | 0.864 | 2.426 |
| INS | 12,219 | 0.243 | 0.223 | 0.0001 | 1.988 | 0.067 | 0.164 | 0.368 |
| TOP | 12,219 | 36.36 | 15.40 | 3.620 | 89.41 | 24.090 | 34.720 | 46.920 |
| INDRATIO | 12,219 | 0.372 | 0.056 | 0.125 | 0.800 | 0.333 | 0.333 | 0.400 |
| So_priv | 12,219 | 0.144 | 0.351 | 0 | 1 | 0 | 0 | 0 |
| MAOW | 12,219 | 0.104 | 0.178 | 0 | 0.891 | 0 | 0.000 | 0.151 |

As can be seen from Table 3, the average value of the investment efficiency of the sample enterprises (Invest_effect) is 4.053, the median is 2.670, the maximum value is 404.7, and the minimum value is 0.0003, which indicates that the difference in investment efficiency among enterprises is large. The average social responsibility information disclosure (CSR) is 0.300, which means that approximately 30% of the samples disclose CSR information. The variance in analysts' attention (ANONO) is 10.770, which indicates that the number of institutions tracking listed companies is not evenly distributed, and there are large differences. In addition, the average value of INDRATIO is 0.372, which indicates that most companies have met the requirements of the "Guiding Opinions on Establishing Independent Director System in Listed Companies" issued in 2001 that the board members of listed companies include at least a 1/3 proportion of independent directors. Descriptive results for the remaining control variables show no serious distribution bias.

### 4.2. Correlation Analysis

Stata12.0 is used to analyze the correlation of variables, and the results are shown in Table 4. It can be seen that investment efficiency and social responsibility information disclosure are significantly negatively correlated at the level of 1%. It shows that when the

company discloses social responsibility information, the value of non-investment efficiency is smaller, and the investment efficiency is higher, which is consistent with our theoretical analysis and also preliminarily verifies our Hypothesis 1: the company that conducts social responsibility information disclosure has higher investment efficiency. And investment efficiency and analyst attention are also significantly negatively correlated at the level of 1%, which provides a preliminary basis for further study of the regulatory role of analyst attention. In addition, the correlation coefficients of independent variables in Table 4 are below 0.3, indicating that there is no serious multiple collinearity problem among the independent variables.

*4.3. Regression Analysis*

Table 5 shows the regression results for Hypothesis 1. Column (1) is the result of using the full sample analysis. The coefficient of the social responsibility information disclosure variable CSR is −1.305, which is significant at the $p < 0.01$ level. Its economic meaning is that the investment efficiency of enterprises disclosing CSR information based on the whole sample situation is about 32% higher than that of non-disclosed enterprises (−1.305 divided by the average of 4.053 of inefficient investment is about 32%), indicating that companies that disclose CSR have better investment efficiency. In column (2), the coefficient of the variable CSR in the context of overinvestment is −2.028, which is significant at the $p < 0.01$ level. Its economic meaning is that the overinvestment level of the company disclosing CSR information is about 41% higher than that of the non-disclosure company (−2.028 divided by the mean of overinvestment 4.900 is about 41%). Column (3) reports the analysis results under the circumstances of insufficient investment. The coefficient of social responsibility disclosure information variable CSR is −0.942, which is also significant at the level of 1%. Its economic meaning is that the underinvestment level of enterprises that disclose CSR information is about 27% higher than that of non-disclosed enterprises (−0.991 divided by the average underinvestment of 3.456 is about 27%), which is consistent with the above theoretical analysis. Enterprises that disclose CSR information have the high loyalty of the stakeholders, attract high-quality employees, benefit from good relations with the market participants, and have a good image and reputation through the transfer of positive signals to the capital market [2]. It may change the company's internal resource allocation and obtain the trust of external investors. The above empirical results support the establishment of hypothesis 1. Companies that disclose CSR information have a higher level of investment efficiency, which is true in both underinvested companies and overinvested companies.

Table 6 shows the test results of Hypothesis 2. This paper uses the median level of the analysts' attention variable to classify the sample into a high analyst attention group and a low analyst attention group. Columns (1) and (2) are the regression results in the full sample context. The coefficient of social responsibility information disclosure variable CSR in the high analyst attention subsample group is only 0.812 (significant at $p < 0.01$), which is lower than that in the low analyst attention group (−1.590, significant at $p < 0.01$), and the difference is −0.521. Also, the Chow test shows that the difference is significant at the level of 5%, indicating that the complementary effect of social responsibility information disclosure is more significant in the case of low attention of analysts, and the effect of improving investment efficiency is more obvious. In the overinvestment samples of columns (3) and (4), the regression results are the same. The difference between the two groups is 1.363, which is significant at the level of 10%. The results in the underinvestment scenarios of columns (5) and (6) are similar. The coefficient of CSR under the low analyst attention group is −1.121 (significant at $p < 0.01$), which is much higher than that of the high analyst attention group (−0.726), and the difference is 0.395, which is also significant at the level of 10%. The above evidence supports the establishment of Hypothesis 2, and social responsibility information disclosure can be used as an effective supplementary channel for incremental information, which has a positive effect on improving the efficiency of corporate investment.

**Table 4.** Correlation analysis of variables.

| | Invest_Effect | CSR | ANONO | Age | LEV | ROA | FRQ | Slack | INS | TOP | INDRATIO | So_priv | MAOW |
|---|---|---|---|---|---|---|---|---|---|---|---|---|---|
| Invest_Effect | 1 | | | | | | | | | | | | |
| CSR | −0.068 *** | 1 | | | | | | | | | | | |
| ANONO | −0.029 *** | 0.200 *** | 1 | | | | | | | | | | |
| Age | −0.024 *** | 0.101 *** | −0.075 *** | 1 | | | | | | | | | |
| LEV | 0.173 *** | 0.183 *** | −0.040 *** | 0.215 *** | 1 | | | | | | | | |
| ROA | 0.027 *** | −0.007 | 0.069 *** | −0.0002 | −0.074 *** | 1 | | | | | | | |
| FRQ | 0.056 *** | −0.034 *** | −0.020 ** | 0.027 *** | 0.052 *** | 0.381 *** | 1 | | | | | | |
| Slack | 0.014 | −0.010 | −0.016 * | 0.031 *** | 0.035 *** | 0.023 ** | 0.031 *** | 1 | | | | | |
| INS | −0.016 * | 0.089 *** | 0.125 *** | 0.153 *** | 0.052 *** | 0.023 ** | 0.010 | 0.010 | 1 | | | | |
| TOP | −0.026 *** | 0.113 *** | 0.050 *** | −0.129 *** | 0.093 *** | 0.011 | 0.009 | −0.008 | 0.051 *** | 1 | | | |
| INDRATIO | 0.005 | 0.037 *** | 0.006 | −0.043 *** | −0.003 | 0.003 | −0.007 | 0.007 | −0.008 | 0.062 *** | 1 | | |
| So_priv | −0.014 | 0.116 *** | 0.009 | 0.058 *** | 0.139 *** | −0.012 | 0.0005 | −0.019 ** | 0.053 *** | 0.126 *** | −0.024 *** | 1 | |
| MAOW | 0.012 | −0.217 *** | 0.032 *** | −0.254 *** | −0.357 *** | 0.027 *** | −0.007 | 0.013 | −0.277 *** | −0.143 *** | 0.072 *** | −0.225 *** | 1 |

Note: ***, **, and * represent at the 1%, 5%, and 10% significance levels, respectively.

**Table 5.** Analysis of Regression Results of CSR Information Disclosure and Investment Efficiency.

| | Invest_Effect | Over_Invest | Under_Invest |
|---|---|---|---|
| | (1) | (2) | (3) |
| CSR | −1.305 *** | −2.028 *** | −0.942 *** |
| | (−8.25) | (−5.79) | (−11.01) |
| ANONO | −0.019 ** | −0.040 ** | −0.030 *** |
| | (−2.82) | (−2.71) | (−8.28) |
| Age | −0.027 * | −0.051 | −0.019 * |
| | (−2.05) | (−1.93) | (−2.48) |
| LEV | 8.880 *** | 13.58 *** | 5.848 *** |
| | (26.28) | (19.11) | (28.70) |
| ROA | 0.492 * | −1.176 | 0.604 *** |
| | (2.46) | (−0.84) | (7.25) |
| FRQ | 0.291 *** | 2.947 *** | 0.247 *** |
| | (3.67) | (4.55) | (7.54) |
| Slack | 0.003 | 0.008 | 0.001 |
| | (1.75) | (1.71) | (1.09) |
| INS | 0.581 | 0.357 | 0.182 |
| | (1.62) | (0.45) | (0.94) |
| TOP | −0.014 ** | −0.028 ** | −0.006 * |
| | (−3.05) | (−2.75) | (−2.24) |
| INDRATIO | 2.530 * | 3.199 | 0.964 |
| | (2.05) | (1.18) | (1.44) |
| So_priv | −0.761 *** | −1.122 * | −0.496 *** |
| | (−3.78) | (−2.46) | (−4.67) |
| MAOW | 2.515 *** | 3.268 ** | 1.147 *** |
| | (5.53) | (3.29) | (4.61) |
| _cons | −0.937 | −0.677 | 0.0432 |
| | (−0.88) | (−0.29) | (0.07) |
| years | control | control | control |
| industry | control | control | control |
| N | 12,219 | 5048 | 7171 |
| *Adj.R*$^2$ | 0.0806 | 0.1067 | 0.1915 |
| F value | 32.51 | 18.71 | 50.91 |

Note: ***, **, and * represent at the 1%, 5%, and 10% significance levels, respectively.

**Table 6.** Regression results based on analyst attention grouping.

| | Invest_Effect | | Over_Invest | | Under_Invest | |
|---|---|---|---|---|---|---|
| | High Analyst Attention (1) | Low Analyst Attention (2) | High Analyst Attention (3) | Low Analyst Attention (4) | High Analyst Attention (5) | Low Analyst Attention (6) |
| CSR | −0.778 *** | −1.590 *** | −1.106 *** | −2.469 ** | −0.726 *** | −1.121 *** |
| | (−5.95) | (−5.21) | (−4.48) | (−3.13) | (−7.20) | (−8.28) |
| Age | −0.014 | −0.039 | −0.029 | −0.057 | −0.021 * | −0.023 * |
| | (−1.29) | (−1.56) | (−1.54) | (−0.92) | (−2.21) | (−2.05) |
| LEV | 1.438 *** | 13.16 *** | 0.372 | 18.71 *** | 4.627 *** | 6.914 *** |
| | (4.03) | (24.21) | (0.53) | (15.27) | (16.48) | (23.66) |
| ROA | 0.908 * | 0.267 | −0.510 | −6.051 * | 2.210 *** | 0.454 *** |
| | (1.97) | (1.00) | (−0.48) | (−2.07) | (7.07) | (4.79) |
| FRQ | 0.286 ** | 0.291 * | 1.138 * | 5.125 *** | 0.182 ** | 0.262 *** |
| | (2.97) | (2.48) | (2.42) | (3.60) | (3.23) | (6.31) |
| Slack | 0.007 | 0.001 | 0.018 | 0.004 | 0.008 ** | −0.0001 |
| | (1.78) | (0.36) | (1.92) | (0.57) | (3.02) | (−0.16) |
| INS | 0.447 | 0.737 | 0.801 | 0.0787 | 0.0121 | 0.275 |
| | (1.44) | (1.12) | (1.39) | (0.05) | (0.05) | (0.95) |

**Table 6.** *Cont.*

| | Invest_Effect | | Over_Invest | | Under_Invest | |
|---|---|---|---|---|---|---|
| | **High Analyst Attention (1)** | **Low Analyst Attention (2)** | **High Analyst Attention (3)** | **Low Analyst Attention (4)** | **High Analyst Attention (5)** | **Low Analyst Attention (6)** |
| TOP | 0.001 | −0.028 *** | 0.001 | −0.054 * | −0.003 | −0.008 * |
| | (0.23) | (−3.35) | (0.15) | (−2.46) | (−0.91) | (−2.17) |
| INDRATIO | 1.378 | 4.481 | 3.162 | 5.326 | −0.735 | 2.657 ** |
| | (1.31) | (1.93) | (1.63) | (0.88) | (−0.89) | (2.62) |
| So_priv | −0.256 | −1.016 ** | −0.148 | −1.695 | −0.391 ** | −0.565 *** |
| | (−1.43) | (−2.80) | (−0.44) | (−1.75) | (−2.88) | (−3.58) |
| MAOW | 1.376 *** | 2.806 ** | 1.769 * | 3.522 | 1.476 *** | 0.524 |
| | (3.54) | (3.25) | (2.43) | (1.62) | (4.89) | (1.35) |
| _cons | 1.816 | −2.811 | 1.454 | −1.994 | 1.626 | −1.280 |
| | (1.28) | (−1.68) | (0.62) | (−0.47) | (1.29) | (−1.72) |
| years | control | control | control | control | control | control |
| industry | control | control | control | control | control | control |
| N | 6378 | 5841 | 3005 | 2043 | 3373 | 3798 |
| $Adj.R^2$ | 0.044 | 0.123 | 0.058 | 0.172 | 0.141 | 0.234 |
| F value | 9.91 | 25.73 | 6.59 | 14.22 | 18.29 | 36.19 |
| Difference between groups | 0.812 ** | | 1.363 * | | 0.395 ** | |
| | (7.35) | | (4.46) | | (6.37) | |

Note: ***, **, and * represent at the 1%, 5%, and 10% significance levels, respectively.

## 5. Robustness Test

### 5.1. Remove Voluntary Disclosure Samples for Regression

Because the voluntary disclosure samples may have the problem of sample self-selection (Zhong & Xu, 2017) [3]. Tables 7 and 8 shown the regression results of the robustness test, excluding the voluntary social responsibility disclosure samples, the sample observation number in the full sample situation and overinvestment and underinvestment situations decreased to 10,836, 4446, and 6390 respectively, and the conclusion remained unchanged after regression.

**Table 7.** Analysis of Regression Results of CSR Information Disclosure and Investment Efficiency (Excluding Voluntary Disclosure Samples).

| | Invest_Effect | Over_Invest | Under_Invest |
|---|---|---|---|
| | **(1)** | **(2)** | **(3)** |
| CSR | −1.899 *** | −2.838 *** | −1.423 *** |
| | (−9.35) | (−6.38) | (−13.14) |
| ANONO | −0.016 * | −0.039 * | −0.027 *** |
| | (−2.13) | (−2.35) | (−6.80) |
| Age | −0.029 * | −0.053 | −0.018 * |
| | (−2.05) | (−1.82) | (−2.19) |
| LEV | 9.571 *** | 14.73 *** | 5.913 *** |
| | (26.00) | (18.84) | (27.31) |
| ROA | 0.363 | −0.891 | 0.430 *** |
| | (1.52) | (−0.48) | (4.48) |
| FRQ | 0.280 *** | 2.659 *** | 0.259 *** |
| | (3.32) | (3.71) | (7.64) |
| Slack | 0.002 | 0.007 | 0.001 |
| | (1.11) | (1.16) | (0.91) |
| INS | 0.735 | 0.449 | 0.261 |
| | (1.84) | (0.51) | (1.24) |

**Table 7.** *Cont.*

|  | Invest_Effect | Over_Invest | Under_Invest |
|---|---|---|---|
|  | **(1)** | **(2)** | **(3)** |
| TOP | −0.015 ** | −0.029 * | −0.006 * |
|  | (−2.82) | (−2.53) | (−2.10) |
| INDRATIO | 2.896 * | 3.838 | 0.832 |
|  | (2.13) | (1.28) | (1.16) |
| So_priv | −0.768 *** | −1.177 * | −0.445 *** |
|  | (−3.43) | (−2.31) | (−3.89) |
| MAOW | 2.469 *** | 3.075 ** | 1.025 *** |
|  | (4.92) | (2.78) | (3.87) |
| _cons | −2.020 | −2.575 | −0.088 |
|  | (−1.75) | (−1.03) | (−0.14) |
| years | control | control | control |
| industry | control | control | control |
| N | 10,836 | 4446 | 6390 |
| Adj.R² | 0.085 | 0.114 | 0.194 |
| F value | 30.71 | 17.86 | 46.11 |

Note: ***, **, and * represent at the 1%, 5%, and 10% significance levels, respectively.

**Table 8.** Regression results based on analyst attention group (excluding voluntary disclosure samples).

|  | Invest_Effect | | Over_Invest | | Under_Invest | |
|---|---|---|---|---|---|---|
|  | **High Analyst Attention (1)** | **Low Analyst Attention (2)** | **High Analyst Attention (3)** | **High Analyst Attention (1)** | **Low Analyst Attention (2)** | **High Analyst Attention (3)** |
| CSR | −1.021 *** | −2.405 *** | −1.061 *** | −1.733 *** | −1.321 *** | −3.604 *** |
|  | (−6.41) | (−5.85) | (−8.50) | (−9.80) | (−4.42) | (−3.41) |
| Age | −0.019 | −0.038 | −0.022 * | −0.021 | −0.032 | −0.049 |
|  | (−1.58) | (−1.37) | (−2.09) | (−1.70) | (−1.63) | (−0.72) |
| LEV | 1.483 *** | 13.93 *** | 4.727 *** | 6.922 *** | 0.165 | 19.59 *** |
|  | (3.81) | (23.80) | (15.57) | (22.67) | (0.21) | (14.72) |
| ROA | 1.216 * | 0.145 | 2.229 *** | 0.282 ** | −0.480 | −6.828 * |
|  | (2.32) | (0.45) | (6.92) | (2.58) | (−0.29) | (−2.05) |
| FRQ | 0.248 * | 0.282 * | 0.173 ** | 0.274 *** | 1.029 * | 4.602 ** |
|  | (2.48) | (2.25) | (3.01) | (6.44) | (2.00) | (2.95) |
| Slack | −0.002 | 0.0004 | 0.007 * | −0.0002 | −0.011 | 0.004 |
|  | (−0.48) | (0.13) | (2.34) | (−0.16) | (−0.88) | (0.43) |
| INS | 0.675 * | 0.893 | 0.252 | 0.308 | 1.045 | 0.147 |
|  | (1.99) | (1.23) | (0.95) | (0.99) | (1.67) | (0.08) |
| TOP | −0.0002 | −0.027 ** | −0.002 | −0.008 * | −0.002 | −0.052 * |
|  | (−0.03) | (−2.87) | (−0.67) | (−1.99) | (−0.27) | (−2.12) |
| INDRATIO | 1.757 | 4.599 | −0.793 | 2.445 * | 4.100 | 6.121 |
|  | (1.55) | (1.80) | (−0.88) | (2.27) | (1.95) | (0.91) |
| So_priv | −0.232 | −1.017 * | −0.326 * | −0.528 ** | −0.150 | −1.707 |
|  | (−1.19) | (−2.52) | (−2.20) | (−3.14) | (−0.41) | (−1.57) |
| MAOW | 1.390 ** | 2.925 ** | 1.475 *** | 0.436 | 1.898 * | 3.465 |
|  | (3.28) | (3.11) | (4.52) | (1.07) | (2.37) | (1.45) |
| _cons | 1.755 | −4.041 * | 1.450 | −1.334 | 1.409 | −4.189 |
|  | (1.18) | (−2.21) | (1.03) | (−1.71) | (0.58) | (−0.89) |
| years | control | control | control | control | control | control |
| industry | control | control | control | control | control | control |

**Table 8.** *Cont.*

| | Invest_Effect | | Over_Invest | | Under_Invest | |
|---|---|---|---|---|---|---|
| | **High Analyst Attention (1)** | **Low Analyst Attention (2)** | **High Analyst Attention (3)** | **High Analyst Attention (1)** | **Low Analyst Attention (2)** | **High Analyst Attention (3)** |
| *N* | 5592 | 5244 | 2958 | 3432 | 2634 | 1812 |
| *Adj.R²* | 0.130 | 0.042 | 0.146 | 0.238 | 0.532 | 0.182 |
| F value | 24.68 | 8.44 | 16.74 | 33.55 | 5.48 | 13.61 |
| Difference between groups | 1.384 *** | | 0.672 *** | | 2.283 *** | |
| | (12.4) | | (13.86) | | (8.28) | |

Note: ***, **, and * represent at the 1%, 5%, and 10% significance levels, respectively.

## 5.2. Replace the Grouping Basis of Analyst Attention

In this round of robustness test (see Table 9), the average value of analyst attention is used as the grouping basis (Lang & Lundholm, 1996) [37], higher than the average is the high analyst attention group, and lower than the mean is the low analyst attention group, and then when we regressed its regulating effect, the conclusion remained unchanged.

**Table 9.** Regression results based on analyst attention grouping (grouped by analyst attention mean).

| | Invest_Effect | | Over_Invest | | Under_Invest | |
|---|---|---|---|---|---|---|
| | **High Analyst Attention (1)** | **Low Analyst Attention (2)** | **High Analyst Attention (3)** | **High Analyst Attention (1)** | **Low Analyst Attention (2)** | **High Analyst Attention (3)** |
| CSR | −0.763 *** | −1.462 *** | −0.779 *** | −1.030 *** | −1.101 *** | −2.365 *** |
| | (−5.00) | (−6.06) | (−7.07) | (−8.81) | (−3.88) | (−4.00) |
| Age | −0.007 | −0.039 | −0.016 | −0.027 ** | −0.020 | −0.065 |
| | (−0.55) | (−1.95) | (−1.54) | (−2.64) | (−0.88) | (−1.47) |
| LEV | 1.328 ** | 11.60 *** | 4.449 *** | 6.679 *** | 0.728 | 17.18 *** |
| | (3.03) | (25.22) | (13.95) | (25.57) | (0.85) | (17.06) |
| ROA | 2.367 ** | 0.312 | 4.282 *** | 0.481 *** | 1.567 | −4.407 * |
| | (3.20) | (1.32) | (9.47) | (5.30) | (0.85) | (−2.28) |
| FRQ | 0.777 *** | 0.259 ** | 0.362 ** | 0.242 *** | 1.556 ** | 3.730 *** |
| | (3.44) | (2.74) | (2.63) | (6.73) | (2.76) | (3.57) |
| Slack | 0.005 | 0.002 | 0.007 * | 0.0002 | 0.032 * | 0.005 |
| | (0.94) | (0.81) | (2.30) | (0.20) | (2.19) | (0.87) |
| INS | 0.367 | 0.663 | 0.140 | 0.133 | 0.684 | −0.080 |
| | (1.00) | (1.26) | (0.53) | (0.52) | (1.01) | (−0.06) |
| TOP | 0.0003 | −0.024 *** | −0.003 | −0.008 * | 0.003 | −0.046 ** |
| | (0.07) | (−3.51) | (−0.91) | (−2.50) | (0.38) | (−2.71) |
| INDRATIO | 2.171 | 3.221 | −0.280 | 1.512 | 3.520 | 4.787 |
| | (1.79) | (1.72) | (−0.31) | (1.69) | (1.58) | (1.04) |
| So_priv | −0.227 | −0.925 ** | −0.328 * | −0.539 *** | −0.237 | −1.376 |
| | (−1.10) | (−3.11) | (−2.20) | (−3.85) | (−0.62) | (−1.81) |
| MAOW | 1.413 ** | 2.494 *** | 1.241 *** | 0.777 * | 2.131 * | 2.843 |
| | (3.11) | (3.63) | (3.78) | (2.30) | (2.53) | (1.73) |
| _cons | 1.064 | −1.756 | 0.831 | −0.567 | 0.641 | −1.111 |
| | (0.44) | (−1.26) | (0.47) | (−0.83) | (0.14) | (−0.34) |
| years | control | control | control | control | control | control |
| industry | control | control | control | control | control | control |

**Table 9.** *Cont.*

| | Invest_Effect | | Over_Invest | | Under_Invest | |
|---|---|---|---|---|---|---|
| | **High Analyst Attention (1)** | **Low Analyst Attention (2)** | **High Analyst Attention (3)** | **High Analyst Attention (1)** | **Low Analyst Attention (2)** | **High Analyst Attention (3)** |
| N | 4726 | 7493 | 2459 | 4712 | 2267 | 2781 |
| $Adj.R^2$ | 0.051 | 0.108 | 0.151 | 0.219 | 0.071 | 0.145 |
| F value | 8.67 | 28.34 | 14.69 | 40.92 | 6.23 | 15.33 |
| Difference between groups | 0.699 ** (6.35) | | 0.251 * (2.82) | | 1.264 ** (5.07) | |

Note: ***, **, and * represent at the 1%, 5%, and 10% significance levels, respectively.

## 6. Conclusions and Prospects

This article makes several theoretical contributions by examining the influence of CSR information disclosure on investment efficiency through analysts' attention, thereby expanding the research perspective on social responsibility information disclosure and investment efficiency. It extends existing stakeholder theory, information asymmetry theory, legalization theory, and neo-institutional theory, providing future research references on CSR, investment efficiency, and analyst attention. Additionally, the paper investigates different mechanisms of analysts' attention in the impact of social responsibility information disclosure on corporate investment efficiency and provides new evidence on how analysts' attention affects micro-enterprise behavior and value. Moreover, this paper contributes to the generation of further research agendas in emerging economies like China, reflecting both the homogeneity and heterogeneity of analyst attention and non-financial reporting in a non-Western context [47].

Furthermore, this article has practical implications. With the implementation of mandatory disclosure policies in state-owned enterprises and publicly listed companies, society's attention to CSR (CSR) in the Chinese market has been increasing, leading to the gradual formation of a market environment [48]. Therefore, the Chinese government should strengthen the establishment and regulation of relevant systems, laws, and regulations to promote CSR information disclosure. Simultaneously, enterprise management should enhance their awareness of fulfilling CSR, actively disclose high-quality social responsibility information, and promote sustainable development. Considering the impact of analysts' attention on the behavior and investment efficiency of listed companies, China should expand the number of analysts with higher supervision capabilities and further improve the governance structure and market information environment of listed companies to enhance their overall value. Regarding the current existence of third-party independent rating agencies that rate social responsibility information reporting, such as Runling Global's RKS rating, regulatory authorities should control its standardization and rationality to encourage its development while also playing a supervisory role.

The limitations of this paper are as follows. First of all, CSR information disclosure is a binary variable based on whether the CSR information is disclosed, and its measurement method needs to be improved. In view of this, a follow-up can be conducted on the measurement method of the variable, which leads to the formation of a more reasonable and effective Chinese social responsibility information disclosure evaluation system. Then, due to the limitation of data collection, this paper mainly focuses on listed companies in Shenzhen and Shanghai, which may have a certain influence on the pertinence of empirical results. In view of this, subsequent studies can be further classified on the sample, such as grouping the whole sample according to industry classification.

In conclusion, we use panel data from listed companies in Shanghai and Shenzhen as a sample to examine the relationship between CSR information disclosure, analyst attention, and investment efficiency in this study. Our key findings indicate that under the current

mandatory disclosure-oriented system, CSR information disclosure can serve as an effective non-financial communication channel, providing additional valuable information to the capital market. It can also be integrated into the corporate governance mechanism to reduce information asymmetry, supervise management, and improve the investment efficiency of enterprises. Additionally, we find that in a more transparent information environment with more constrained managers, the impact of CSR disclosure on investment efficiency is weaker when analyst attention is higher. Conversely, the impact of CSR information disclosure on investment efficiency is more pronounced when analyst attention is low.

**Author Contributions:** Conceptualization, Z.L.; Software, Z.L.; Validation, Y.L.; Investigation, Z.L.; Data curation, S.L.; Writing—original draft, S.L. and Y.L.; Writing—review & editing, Z.H.; Visualization, H.Z.; Supervision, H.Z.; Project administration, Z.H.; Funding acquisition, Z.H. All authors have read and agreed to the published version of the manuscript.

**Funding:** This research received no external funding.

**Institutional Review Board Statement:** Not applicable.

**Informed Consent Statement:** Not applicable.

**Data Availability Statement:** Not applicable.

**Conflicts of Interest:** The authors declare no conflict of interest.

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
