# Peer review of "Does CSR Information Disclosure Improve Investment Efficiency? The Moderating Role of Analyst Attention"

_sustainability, doi:10.3390/su151612310_

Round 1
Reviewer 1 Report
The strengths of the study are its relevance, the logical presentation of materials, an exciting idea is the basis of the article. The set research hypotheses and the presented logical framework for the paper always attract more attention and simplify the perception of the study.
Yet, despite the merits, the presentation of the study could be better in the following areas:
1) I propose to change the name. The current is more about keywords and does not reflect any connection.
2) there is a big gap in the analysis of the literature, especially in the newest one (2023), for example, “The existing literature pays more attention to the economic consequences of CSR information disclosure (Ullmann, 1985)” – it is not good the ground on such “senior” literature, for example, this one “Dudek M. Methodology for assessment of inclusive social responsibility of the energy industry enterprises” much more appropriate concerning comparison. For example, in this case “This paper adopts stakeholder theory (Freeman, 1999), information asymmetry theory (Stiglitz, 1974) and organizational legitimacy theory (Suchman, 1995), and refer to Richardson's (2006)…” it is completely ok to use this references because they are the base. But it is needed to “refresh” others. Malynovska Y. Enhancing the Activity of Employees of the Communication Department of an Energy Sector Company – maybe this one would be useful concerning enterprises and/or employees.
3) Maybe it would be helpful for readers if the authors add the research questions to the existing hypotheses.
4) Clarify the methodology. In the current way, it is rather chaotic. Maybe because of the whole article’s design – it is awful and complicates the reading of the study. However it is not a problem at this stage, cause the editing department will fix it for sure.
5) I suggest dividing the current “6. Conclusions and prospects” into 3 sections: 1) Discussion - comparison with existing studies, in which it is necessary to emphasize the novelty of the study and its practical/theoretical/methodological contribution and highlighting how the author's research differs from them. 2) Limitations of the study; and 3) Conclusions – the overall conclusions or the study, if possible with the ways of further research.
Author Response
The strengths of the study are its relevance, the logical presentation of materials, an exciting idea is the basis of the article. The set research hypotheses and the presented logical framework for the paper always attract more attention and simplify the perception of the study.
Yet, despite the merits, the presentation of the study could be better in the following areas.
Authors’ Responses:
Thank you very much for your kind words. We are very encouraged by your comments. We have revised the manuscript based on your specific comments. If there are any other modifications we could make, we would like very much to modify them and we appreciate your help.
- I Propose to change the name. The current is more about keywords and does not reflect any connection.
Authors’ Responses:
Thank you very much for providing valuable feedback. We agree that the original title used does not adequately reflect the internal logic of the study and is only relevant to the keywords. So we replace the title with: Does corporate social responsibility information disclosure improve investment efficiency?The moderating role of analyst attention.
Please see the red text on Page 1.
- There is a big gap in the analysis of the literature, especially in the newest one (2023), for example, “The existing literature pays more attention to the economic consequences of CSR information disclosure (Ullmann, 1985)” – it is not good the ground on such “senior” literature, for example, this one “Dudek M. Methodology for assessment of inclusive social responsibility of the energy industry enterprises” much more appropriate concerning comparison. For example, in this case “This paper adopts stakeholder theory (Freeman, 1999), information asymmetry theory (Stiglitz, 1974) and organizational legitimacy theory (Suchman, 1995), and refer to Richardson's (2006)…” it is completely ok to use this references because they are the base. But it is needed to “refresh” others. Malynovska Y. Enhancing the Activity of Employees of the Communication Department of an Energy Sector Company – maybe this one would be useful concerning enterprises and/or employees.
Authors’ Responses:
Thank you very much for your feedback. We agree that the literature previously used in the paper is not novel enough for the point to be fully substantiated. We replaced the document "Dudek M. Methodology for assessment of inclusive social responsibility of the energy industry enterprises", so as to reduce the gap of literature analysis in this section. We have made certain adjustments to the literature analysis, and have included some basic literature in the theoretical analysis, adding more up-to-date research literature
Please see the red text on Page 3.
References:
Jiang, Y., Wang, C., Li, S., & Wan, J. (2022). Do institutional investors' corporate site visits improve ESG performance? Evidence from China. Pacific-Basin Finance Journal, 76, 101884.
Rashid, A., Shams, S., Bose, S., & Khan, H. (2020). CEO power and CSR (CSR) disclosure: does stakeholder influence matter?. Managerial Auditing Journal, 35(9), 1279-1312.
Dudek, M., Bashynska, I., Filyppova, S., Yermak, S., & Cichoń, D. (2023). Methodology for assessment of inclusive social responsibility of the energy industry enterprises. Journal of Cleaner Production, 394, 136317.
- Maybe it would be helpful for readers if the authors add the research questions to the existing hypotheses.
Authors’ Responses:
Thank you very much for providing valuable feedback. We agree with this point and acknowledge that adding research questions to the existing hypotheses would indeed help readers better understand our study. We will explicitly state the research questions in the final part of the "Introduction" section and ensure they align with the existing hypotheses. We believe that this improvement will further enhance the quality and readability of our research.
Please see the red text on Page 3.
- Clarify the methodology. In the current way, it is rather chaotic. Maybe because of the whole article’s design – it is awful and complicates the reading of the study. However it is not a problem at this stage, cause the editing department will fix it for sure.
Authors’ Responses:
Thank you for your feedback on the methodology section. We understand your concern about the current presentation of the methodology and how it may be confusing and difficult to comprehend. We apologize for any inconvenience this may have caused in understanding the overall design of the article. We have made modifications to the title of our research design based on the reference from your journal.
Please see the red text on Page 7,8, and 10.
- I suggest dividing the current “6. Conclusions and prospects” into 3 sections: 1) Discussion - comparison with existing studies, in which it is necessary to emphasize the novelty of the study and its practical/theoretical/methodological contribution and highlighting how the author's research differs from them. 2) Limitations of the study; and 3) Conclusions – the overall conclusions or the study, if possible with the ways of further research.
Authors’ Responses:
Thank you very much for your suggestion to divide the "6. Conclusion and Outlook" section into three parts. We greatly appreciate your input on improving the structure and clarity of the article. Based on your recommendation, we believe the "Conclusion and Outlook" section can be reorganized as follows: (1) Discussion: In this part, we will compare our research to existing studies, emphasizing the theoretical and practical contributions of our research and further highlighting its significance and impact. (2) Limitations of the study: This section will provide an honest assessment of the limitations in our variable measurement or data itself. We will discuss potential biases, constraints, or sources of errors that may affect our research findings and suggest possible avenues for future improvement. (3) Conclusion and future research: In this part, we will summarize the overall conclusions of the study, encapsulating key insights and findings. We believe that this reorganization will enhance the clarity and coherence of the article, making it easier for readers to understand the research process and its significance.
Please see the red text on Page 21-22.

Reviewer 2 Report
The problem of the correlation between corporate social responsibility information disclosure, corporate investment efficiency, and the role of analysts in their interrelation which is described in the article is relevant. The topic is actual and of interest. The title is clear and appropriate to the paper’s subject matter.
The text is written clearly, concisely, stylistically, and technically correct. Actual scientific material, logically and reasonably presented. The literature used makes it possible to reveal the degree of knowledge of the problem and highlight issues that require further study. Every reference cited in the text is also present in the reference list.
The data analysis is adequate and allows for achieving the purpose of the study. The scientific argumentation of the main provisions of the article is logical and convincing. Conclusions are correctly and logically derived from the evidence and arguments presented data.
Below are several suggestions that I hope will be helpful in the paper:1. The abstract is not focused and does not fully and clearly express the goal. The authors of this paper should add more details about the theoretical and practical contributions. The statement in the abstract such as "The above empirical results show that the social responsibility information disclosed under ...." is not logical and should be corrected.
2. The units of measure in Table 1 should be specified.
3. Some tables (tables 2, 3, 4, 6, 7, 8, 9) in the article are not formatted correctly, which led to their break on different pages. Authors should pay attention to this issue and make appropriate tables design changes to ensure continuous reading and data analysis.
The language used is academic in nature, which is appropriate for a research paper. Overall, the quality of English language is sufficient, showcasing a proficient level of writing and effectively conveying the main ideas.
Author Response
Responses to Reviewer 2’s Comments
The problem of the correlation between corporate social responsibility information disclosure, corporate investment efficiency, and the role of analysts in their interrelation which is described in the article is relevant. The topic is actual and of interest. The title is clear and appropriate to the paper’s subject matter.
The text is written clearly, concisely, stylistically, and technically correct. Actual scientific material, logically and reasonably presented. The literature used makes it possible to reveal the degree of knowledge of the problem and highlight issues that require further study. Every reference cited in the text is also present in the reference list.
The data analysis is adequate and allows for achieving the purpose of the study. The scientific argumentation of the main provisions of the article is logical and convincing. Conclusions are correctly and logically derived from the evidence and arguments presented data.
Authors’ Responses:
Thank you very much for your kind words. We are very encouraged by your comments.
- The abstract is not focused and does not fully and clearly express the goal. The authors of this paper should add more details about the theoretical and practical contributions. The statement in the abstract such as "The above empirical results show that the social responsibility information disclosed under ...." is not logical and should be corrected.
Authors’ Responses:
Thank you for your feedback. We have supplemented the abstract with details of theoretical and practical contributions. Also we modified " The above empirical results show that the social responsibility information disclosed under...." part so that it can be more logical and can be better understood.
Please see the red text on Page 1.
- The units of measure in Table 1 should be specified.
Authors’ Responses:
Thank you for your feedback. We apologize for the oversight in failing to specify the units of measure in Table 1. We have now revised the document to include the appropriate units. Please find the updated version below:
Table 1. Sample Screening(number of companies)
|
Screening process |
2013 |
2014 |
2015 |
2016 |
2017 |
2018 |
2019 |
Total |
|
Sample of A-share listed companies that disclosed investment amount in 2013-2019 |
2341 |
2470 |
2515 |
2632 |
2823 |
3118 |
3488 |
19387 |
|
Less: Sample of listed companies with missing investment efficiency model data |
256 |
385 |
191 |
174 |
319 |
499 |
679 |
2503 |
|
Sample of listed companies in the financial and insurance industry |
135 |
135 |
143 |
144 |
144 |
146 |
148 |
995 |
|
ST, PT listed company sample |
10 |
17 |
14 |
21 |
13 |
11 |
5 |
91 |
|
Sample of listed companies without analyst forecasts |
339 |
324 |
480 |
619 |
583 |
512 |
455 |
3312 |
|
Sample of listed companies with missing financial data |
112 |
38 |
27 |
64 |
20 |
4 |
2 |
267 |
|
Final sample |
1489 |
1571 |
1660 |
1610 |
1744 |
1946 |
2199 |
12219 |
Once again, we apologize for any confusion caused and appreciate your diligence in reviewing our work.
- Some tables (tables 2, 3, 4, 6, 7, 8, 9) in the article are not formatted correctly, which led to their break on different pages. Authors should pay attention to this issue and make appropriate tables design changes to ensure continuous reading and data analysis.
Authors’ Responses:
Thank you for bringing this issue to our attention. We apologize for the incorrect formatting of tables 2, 3, 4, 6, 7, 8, and 9, which caused them to break on different pages. We understand the importance of ensuring continuous reading and data analysis for a seamless reader experience. We have carefully reviewed the formatting of these tables and made the necessary design changes to address the issue. By adjusting the column widths and page breaks, we have now ensured that the tables will be presented in a continuous manner without any breaks on different pages. We apologize for any inconvenience caused by the initial formatting error and appreciate your feedback.

Reviewer 3 Report
Thank you for giving me an opportunity to review.
The study abstract presents a clear and focused picture that examines an important topic related to CSR and investment efficiency in China.
In the introduction section, the inclusion of excessive details and references to previous research can make it challenging for readers to grasp the main objective of the study. Revising the introduction to provide a clearer and more concise description of the research topic, research gap, and objectives would improve its effectiveness.
H1: The hypothesis lies in its theoretical foundation and recognition of the importance of CSR information disclosure in improving investment efficiency. However, addressing the demerits by providing clear definitions, accounting for limitations, and stating the direction of the relationship would strengthen the hypothesis further.
H2: The hypothesis does not explicitly address the potential mechanisms through which low analyst attention enhances the effect of CSR disclosure on investment efficiency. Discussing these mechanisms or providing potential explanations would add depth to the hypothesis.
Moreover, the references need to be updated with more latest years with A-ranked journals in LR section.
I am unable to find the discussion section which was highlighted in Introduction section. However, the conclusion highlights the theoretical contributions of the study in expanding the research perspective of CSR disclosure and investment efficiency, as well as the practical implications for the Chinese government and enterprises.
Mentioned in given options.
Author Response
Responses to Reviewer 3’s Comments
Thank you for giving me an opportunity to review.
The study abstract presents a clear and focused picture that examines an important topic related to CSR and investment efficiency in China.
Authors’ Responses:
Thank you very much for your kind words. Your expertise and valuable suggestions have played a pivotal role in the improvement of our paper.
- In the introduction section, the inclusion of excessive details and references to previous research can make it challenging for readers to grasp the main objective of the study. Revising the introduction to provide a clearer and more concise description of the research topic, research gap, and objectives would improve its effectiveness.
Authors’ Responses:
Thank you very much for your feedback. In the part of introduction, we have simplified the expression of the text and deleted some references, so that our article can more highlight the research content and the contribution of this research to related fields
Please see the red text on Page 2,3.
- H1: The hypothesis lies in its theoretical foundation and recognition of the importance of CSR information disclosure in improving investment efficiency. However, addressing the demerits by providing clear definitions, accounting for limitations, and stating the direction of the relationship would strengthen the hypothesis further.
Authors’ Responses:
Thank you very much for your specific feedback. In response to your suggestions, we will provide further details on the following aspects:
Firstly, we will provide a clearer definition to ensure readers have a more accurate understanding of the relationship between CSR information disclosure and investment efficiency. We will explain the concept of CSR information disclosure and its operational definition in our research, so that readers can understand the scope and purpose of our study more accurately. Please see the red text on Page 1.
Secondly, we will carefully consider the limitations of our research and clearly state them in the sixth chapter of the paper. While our research framework and theoretical basis can support our hypotheses, we also acknowledge that there may be some inherent limitations in the study. We will discuss these limitations honestly and offer suggestions for future research that can address them more effectively. Please see the red text on Page 21,22.
Lastly, we will depict the direction of the relationship between CSR information disclosure and investment efficiency through Figure 1. This will help readers to have a more intuitive understanding and interpretation of our research results. Please see the red text on Page 7.
In conclusion, we appreciate your review of our paper, and we will carefully consider your suggestions and make appropriate revisions to strengthen our hypotheses. We believe that through these improvements, our research will become more accurate and reliable.
- H2: The hypothesis does not explicitly address the potential mechanisms through which low analyst attention enhances the effect of CSR disclosure on investment efficiency. Discussing these mechanisms or providing potential explanations would add depth to the hypothesis.
Authors’ Responses:
Thank you for the valuable feedback on our hypotheses. Due to the constraints of our research methodology, we have not provided a more detailed mechanism analysis of the impact of analyst attention on the main effect. However, before proposing Hypothesis 2, we have provided a detailed explanation of how analyst attention influences the main effect through the analysis of information asymmetry theory and stakeholder theory. We appreciate your suggestion.
- I am unable to find the discussion section which was highlighted in Introduction section. However, the conclusion highlights the theoretical contributions of the study in expanding the research perspective of CSR disclosure and investment efficiency, as well as the practical implications for the Chinese government and enterprises.
Authors’ Responses:
Thank you very much for pointing out the missing discussion section. We apologize for any confusion caused. In our introduction, we intended to analyse the current state of literature both domestically and internationally, further examining potential research limitations, and introducing the research hypotheses for the subsequent chapters. Therefore, a more specific discussion on the theoretical contributions of this study from the perspective of expanding corporate social responsibility disclosure and investment efficiency research, as well as its practical implications for the Chinese government and enterprises, has been allocated to Chapter 6.
- Moreover, the references need to be updated with more latest years with A-ranked journals in LR section.
Authors’ Responses:
Thank you very much for your feedback. We have replaced some of the literature to make it more recent, and they come from higher-level journals, so that they can better support our ideas in the literature review section.
Please see the red text on Page 22 and 23.
References:
Jiang, Y., Wang, C., Li, S., & Wan, J. (2022). Do institutional investors' corporate site visits improve ESG performance? Evidence from China. Pacific-Basin Finance Journal, 76, 101884.
Rashid, A., Shams, S., Bose, S., & Khan, H. (2020). CEO power and CSR (CSR) disclosure: does stakeholder influence matter?. Managerial Auditing Journal, 35(9), 1279-1312.
Dudek, M., Bashynska, I., Filyppova, S., Yermak, S., & Cichoń, D. (2023). Methodology for assessment of inclusive social responsibility of the energy industry enterprises. Journal of Cleaner Production, 394, 136317.
Thank you for your feedback. We will revise the introduction section again. Thank you once again for your valuable input.

Round 2
Reviewer 1 Report
I believe that the authors successfully coped with the task of correcting the manuscript so that it could be printed.